# Stimulation Montage Achieves Balanced Focality and Intensity

Yushan Wang [1] , Jonathan Brand [1] and Wentai Liu [1,2,3,4,*]

1  Biomimetic Research Lab, Department of Bioengineering, University of California, Los Angeles, Los Angeles, CA 90095, USA; yushanwang@g.ucla.edu (Y.W.); jdbrand@g.ucla.edu (J.B.)
2  Department of Electrical and Computer Engineering, University of California, Los Angeles, Los Angeles, CA 90095, USA
3  California NanoSystems Institute, University of California, Los Angeles, Los Angeles, CA 90095, USA
4  Brain Research Institute, University of California, Los Angeles, Los Angeles, CA 90095, USA
*  Correspondence: wentai@ucla.edu

**Abstract:** Transcranial direct current stimulation (tDCS) is a non-invasive neuromodulation technique to treat brain disorders by using a constant, low current to stimulate targeted cortex regions. Compared to the conventional tDCS that uses two large pad electrodes, multiple electrode tDCS has recently received more attention. It is able to achieve better stimulation performance in terms of stimulation intensity and focality. In this paper, we first establish a computational model of tDCS, and then propose a novel optimization algorithm using a regularization matrix $\lambda$ to explore the balance between stimulation intensity and focality. The simulation study is designed such that the performance of state-of-the-art algorithms and the proposed algorithm can be compared via quantitative evaluation. The results show that the proposed algorithm not only achieves desired intensity, but also smaller target error and better focality. Robustness analysis indicates that the results are stable within the ranges of scalp and cerebrospinal fluid (CSF) conductivities, while the skull conductivity is most sensitive and should be carefully considered in real clinical applications.

**Keywords:** transcranial direct current stimulation (tDCS); optimization model; quantitative evaluation metrics; conductivity; robustness test

## 1. Introduction

As a non-invasive neuromodulation method, transcranial direct current stimulation (tDCS) shows therapeutic potential to treat many brain disorders and improve brain functions, such as major depression [1–3], epilepsy [4–6], and Parkinson's disease [7–9]. It has also garnered great interest because it may benefit healthy individuals as well [10–13]. Conventional tDCS applies a constant, low direct current through two large pad electrodes to stimulate a specific brain region. It is challenging to achieve precise activation or inhibition at a specific region without interfering with other regions of the brain. Numerous efforts have been made to improve this promising technique, such as using multiple electrodes to achieve focalized EEG-guided stimulation. For example, GTEN system (EGI, Eugene, OR, USA) with 256 channels integrated the reciprocity theorem and EEG-based source localization tools such as sLORETA. These reciprocity principle based methods use observed EEG patterns as a guide to maximize the directional electric field at the target. However, they have many deficiencies. First of all, updated source localization algorithms such as gFOTV can improve localization accuracy and degree of focalization [14,15]. In addition, the density of the electrode system does affect stimulation outcomes, and 256 may not be high enough to reach the plateau [16–19]. The most important drawback is that the reciprocity theorem may not be ideal, especially in the multiple target case. It may stimulate the average location of these targets, and results in a broadly distributed stimulation pattern. Consequently, one of the emerging challenges of utilizing the high-density electrode system for tDCS is to determine the optimal current at each electrode. An optimal current pattern

will enable multi-electrode systems to provide stimulation with high focality, accuracy, and intensity.

So far, optimization-based methods are the most popular solution to assign injected current values. As an overdetermined problem, the Least Squares solution is a straightforward approach [20], which minimizes the second order error term. Considering the safety issue in clinical use, constraints are then added when applying the algorithm; thus, it can be referred to as the Constrained Least Squares method (CLS). Its optimal current pattern often produces relatively focal stimulation but low stimulation intensity; one possible explanation for this is that the target region is generally tiny compared to the whole brain. To overcome this challenge, the Least Squares method has been improved by assigning weight to balance the tiny target region and large non-target ones, in a technique called the Weighted Least Squares method [20,21]. It is able to produce higher stimulation intensity; however, the weight factor must be given by the clinician, which is a non-trivial task. Another possible solution is to change the L2-norm of the error vector to an L1-norm approach. The L1-norm is applied to achieve more focal stimulation because the fidelity term based on the L1-norm is more robust and results in a non-uniform error distribution, which we have reported previously [22]. However, itis very computationally expensive. Because of its computational drawbacks, we do not explore the effects of using the L1-norm in this investigation. Another developed technique optimizes for the intensity at the target region, and is named the Max Intensity (MI) method [17,23]. This method tends to achieve high stimulation intensity, but is more likely to activate large non-target areas. Thus, this motivates the introduction of more constraints on non-target areas to improve the MI method [24,25]. However, the additional constraints may lead to situations of no feasible solution set and, also, longer computational time. These improvements all indicate the key point behind this optimization problem—that is, to find ways of balancing the stimulation intensity and focality. One novel solution named Linearly Constrained Minimum Variance (LCMV) [20] adopts ideas from the beamforming problem. The algorithm has a hard constraint in that the stimulation intensity at the target region is enforced to be exactly equal to the desired one, while a cost function minimizes the energy of the non-target regions. This strategy is aimed to ensure the stimulation effectiveness, but has pitfalls. LCMV minimizes the effects on the non-target areas under the premise that the hard constraint is fulfilled. When the desired electric field at the target regions is difficult to achieve, it can greatly sacrifice the non-target region and produce a spread-out electric field distribution. In the worst case, it may even fail to attain the hard constraint, thus giving no feasible solution.

To balance the trade-off and overcome these pitfalls, we propose a new method combining the principles of LCMV and MI. The new method, *Stimulation with Balanced Focality and Intensity (SBFI)*, maximizes the energy in the target region and minimizes the rest of the energy in non-target regions. In addition, we also adopt an idea from the Weighted Least Squares method by adding a stimulation parameter $\lambda$ in SBFI to balance the intensity and focality of the target regions and non-target regions. Computational simulation experiments were conducted using the aforementioned methods and the proposed SBFI method. The quantitative results show that SBFI achieves better performance in balancing the stimulation intensity and focality for both single and multiple targets studies. Robustness experiments indicate that the results are stable with different scalp and CSF conductivities, while skull conductivity is most sensitive and should be carefully considered in real clinical applications. The proposed optimization method SBFI shows a good robustness among tested methods in terms of the overall electric field distribution deviations and the maximum intensity changes at the target area.

## 2. Materials and Methods

### 2.1. Framework and Computational Model

To formulate this optimization problem, we consider the head as a volume conduction model, which consists of multiple tissues, each with different electrical conductivity. Furthermore, the cortex is discretized into $n$ elements, and the electric field in the cortex

is denoted by $e_{3n \times 1}$. The realistic head model in this study is derived from a FieldTrip template [26], which provides anatomical information of the scalp, skull, CSF, and cortex. The conductivity values are adopted from the literature [20], where $\sigma_{Scalp} = 0.465$ S/m, $\sigma_{Skull} = 0.01$ S/m, $\sigma_{CSF} = 1.65$ S/m, and $\sigma_{Cortex} = 0.2$ S/m.

For a stimulation system with $m$ electrodes, we use $s_{m \times 1}$ to denote the injected current in the system. In this study, we choose $m = 342$; thus, $s$ has a dimension of $342 \times 1$. This high-density electrode system, with more degrees of freedom, is able to provide a better stimulation pattern than a low-density system [18]. The electrode location is based on the international electroencephalography (EEG) system. The electrode model was constructed using SolidWorks (Dassault Systèmes SOLIDWORKS Corp., Waltham, MA, USA). To simulate real clinical conditions, the electrode has both a metal layer and gel layer.

Considering the fact that head tissues are mainly resistive when tDCS is applied, the electric field distribution can be regarded as quasi-static. Under this condition, the applied current $s_{m \times 1}$ and electric field $e_{3n \times 1}$ are linearly related as $e_{3n \times 1} = K_{3n \times m} \cdot s_{m \times 1}$. The coefficient matrix $K_{3n \times m}$, known as the lead field matrix, provides the mapping information between the injected currents of the electrode system, and the electric field value at each voxel of the brain. $K_{3n \times m}$ is obtained by FEM and solving the Laplace equation in COMSOL Multiphysics (COMSOL Inc., Burlington, MA, USA).

### 2.2. Optimization Model

With the intent to balance the tradeoff between stimulation focality and intensity, Stimulation with Balanced Focality and Intensity (SBFI) combines both in the cost function. The focality is represented by the total energy of the non-target regions $\| Ds \|^2$. Here, $D$ is the submatrix of $K$ relating the injected current at non-target regions. The intensity can be expressed by $e_0{}^T Cs$. The distribution of the desired electric field intensity at the target is $e_0$. $C$, as the submatrix of $K$, is the coefficient matrix of current at the target region (s). Thus, the cost function of SBFI can be further written as:

$$s = arg\min_{s} \left( \frac{n_{tar}}{n_{non}} \| Ds \|^2 - \lambda e_0{}^T Cs \right) \qquad (1)$$

where $\lambda$ is the optimization parameter to balance the first term of focality and the second term of intensity. Higher $\lambda$ favors intensity, while smaller $\lambda$ tends to have better focality; in this work, $\lambda$ is chosen empirically based on parameter sweep simulations. The number of voxels at the target and non-target regions are $n_{tar}$ and $n_{non}$, respectively. This convex optimization problem can be solved efficiently by software such as CVX, and the only unknown current pattern $s$ is therefore obtained.

Furthermore, the algorithm can be expanded to fulfil the needs of targeting multiple brain regions in a single stimulation session. For example, different erectile dysfunction (ED) symptoms are mapped to different cortical targets, and stimulating a single target may be insufficient to address multi-dimensional ED pathology [27–29]. In addition, network-targeted transcranial direct current stimulation (net-tDCS) is able to change the excitability of the sensorimotor network, and show the potential to manipulate network connectivity patterns [30]. Similarly, multi-target stimulation is desirable in the potential application of stopping seizure with the guidance of neural recording techniques [31,32]. However, stimulation effects at different targets may not be the same if complex brain structure and other factors are considered. Thus, the optimization parameter $\lambda$ is further refined by $\lambda = diag(\lambda_1, \lambda_1, \ldots, \lambda_{n_{tar}})$ to balance the stimulation effects of different targets.

$$s = arg\min_{s} \left( \frac{n_{tar}}{n_{non}} \| Ds \|^2 - e_0{}^T \lambda^T Cs \right) \qquad (2)$$

### 2.3. Safety Constraints

To guarantee the stimulation is within the safety limits, it is necessary to introduce constraints for the optimization model. Generally, there are three common safety concerns.

First, the sum of all current inflow should be equal to the sum of all current outflow based on the charge conservation law.

$$\sum s_i = 0 \tag{3}$$

Second, the current injected into each electrode cannot exceed $I_{max}$. This is especially important for high-density electrode systems to avoid side effects such as pain and skin injury. This constraint can be written as:

$$|s_i| \le I_{max}, \text{ for any } i \tag{4}$$

Lastly, we limit the sum of all current inflow to the body. If we use $I_{total}$ to represent the maximum sum currents injected into the body, then the constraint will be:

$$\sum |s_i| \le 2I_{total} \tag{5}$$

In this study, we set $I_{max} = 2$ mA, and $I_{total} = 4$ mA, which are reported to be safe [33–35].

The feasible sets of the optimization model are defined as:

$$S = \left\{ s \in R^m : \sum s_i = 0, \ |s_i| \le I_{max}, \ \sum |s_i| \le 2I_{total} \right\} \tag{6}$$

### 2.4. Experiment Design

To test the performance of the proposed algorithm SBFI, and compare it to the conventional two-electrode system and some available algorithms mentioned, we designed various stimulation cases based on different clinical applications. First, the simplest test is to stimulate a single target containing only one voxel in the motor cortex with a desired intensity of 0.3 V/m (Figure 1), i.e., $n_{tar} = 1$, $e_0$ has a dimension of $3 \times 1$, and $C$ has a dimension of $3 \times m$; this scenario mimics common clinical use. The second test is to stimulate multiple targets derived from synthetic data [15]. As indicated in Figure 2, the stimulation targets contain three brain regions with a maximum electric field of 0.3727 V/m in the left frontal lobe (region 1), 0.3522 V/m in the left occipital lobe (region 2), and 0.2841 V/m in the left temporal lobe (region 3). In this case, target voxels refer to those with an electric field intensity larger than 0.05 V/m. Therefore, $n_{tar} = 111$, $e_0$ has a dimension of $333 \times 1$, and $C$ has a dimension of $333 \times m$. Our third test measures how each algorithm performs with real EEG data from a seizure patient. The target regions are identified by EEG source localization [15,36]. In addition to examining SBFI performance on multiple targets in real seizure data, this study also demonstrates the applicability of our method to EEG-guided brain stimulation. Similar to the synthetic case, there are three target areas in the EEG-guided case, shown in Figure 3. One is in the occipital cortex (region 1) with maximum electric field of 0.1107 V/m, while the other two are on the left and right prefrontal lobe (region 2 and 3) with intensities of 0.0702 V/m and 0.0844 V/m, respectively. Due to the infeasibility of a conventional two-electrode system to target multiple areas, it is excluded in the performance comparison study of optimization algorithms for multiple targets. The comparison is therefore between SBFI, CLS, and MI.

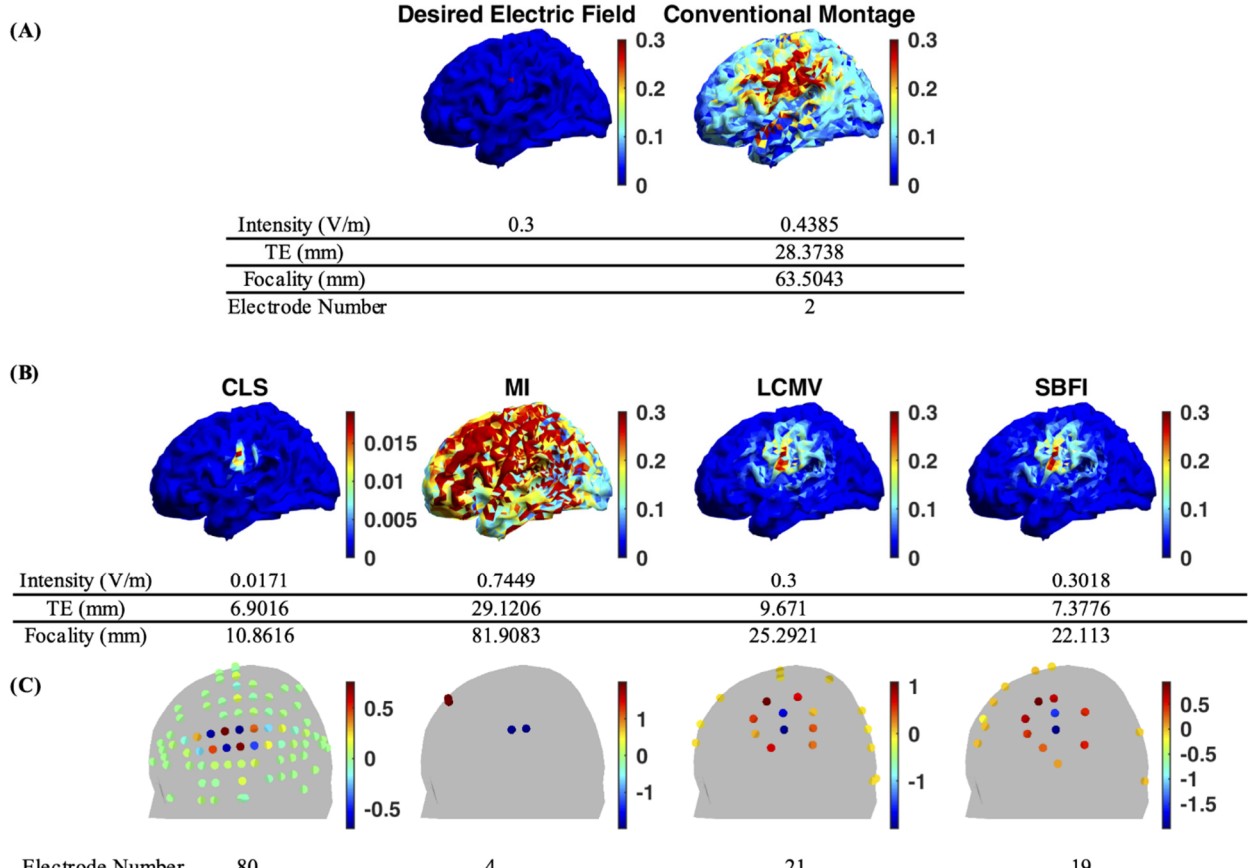

**Figure 1.** Results for the single target protocol. (**A**) Desired electric field and result of conventional montage. (**B**) Results of different optimization algorithms. The color scale represents intensity of the electric field, in V/m. (Note: Results are shown at the same scale [0, 0.3] V/m for various algorithms except CLS. The intensity above 0.3 V/m is saturated for visualization.) (**C**) The electrode configurations calculated by different algorithms. The Conventional montage and MI methods produce spread-out electric field distributions with a small number of electrodes. The CLS method can achieve focal stimulation, but the intensity is too weak. In addition, it requires the utilization of 80 electrodes, which is another drawback compared to the others. Both LCMV and SBFI successfully achieve focusing stimulation with sufficient intensity. SBFI is slightly better than LCMV in terms of intensity, target error, and focality.

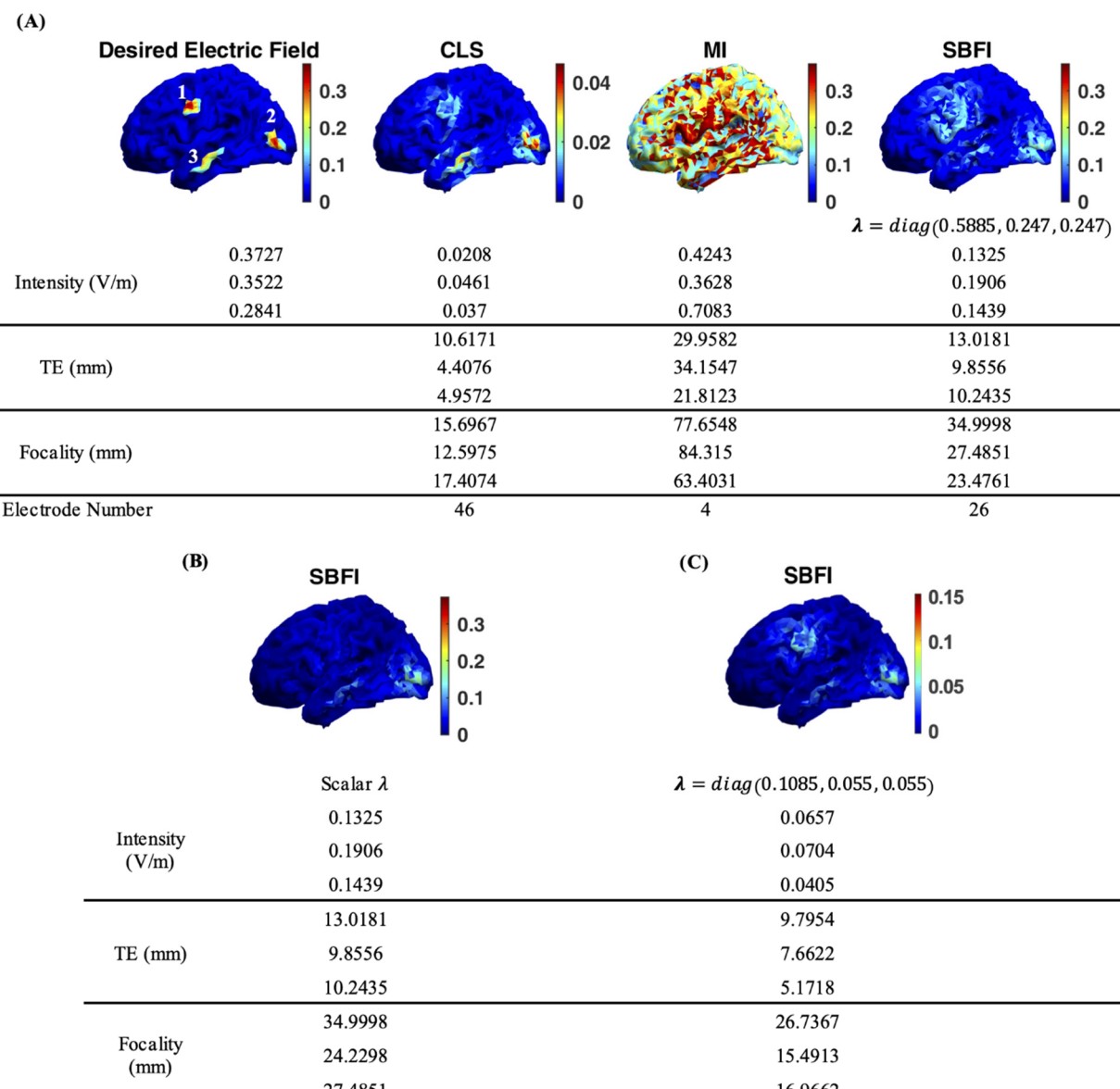

| | Desired Electric Field | CLS | MI | SBFI |
|---|---|---|---|---|
| | | | | $\lambda = diag(0.5885, 0.247, 0.247)$ |
| Intensity (V/m) | 0.3727 | 0.0208 | 0.4243 | 0.1325 |
| | 0.3522 | 0.0461 | 0.3628 | 0.1906 |
| | 0.2841 | 0.037 | 0.7083 | 0.1439 |
| TE (mm) | | 10.6171 | 29.9582 | 13.0181 |
| | | 4.4076 | 34.1547 | 9.8556 |
| | | 4.9572 | 21.8123 | 10.2435 |
| Focality (mm) | | 15.6967 | 77.6548 | 34.9998 |
| | | 12.5975 | 84.315 | 27.4851 |
| | | 17.4074 | 63.4031 | 23.4761 |
| Electrode Number | | 46 | 4 | 26 |

| | SBFI (B) | SBFI (C) |
|---|---|---|
| | Scalar $\lambda$ | $\lambda = diag(0.1085, 0.055, 0.055)$ |
| Intensity (V/m) | 0.1325 | 0.0657 |
| | 0.1906 | 0.0704 |
| | 0.1439 | 0.0405 |
| TE (mm) | 13.0181 | 9.7954 |
| | 9.8556 | 7.6622 |
| | 10.2435 | 5.1718 |
| Focality (mm) | 34.9998 | 26.7367 |
| | 24.2298 | 15.4913 |
| | 27.4851 | 16.9662 |

**Figure 2.** Results for the synthetic multiple targets protocol. (**A**) Results of different optimization algorithms for the synthetic multiple targets protocol. The color scale represents intensity of the electric field, in V/m. (Note: Results are shown at the same scale [0, 0.3727] V/m for various algorithms except for CLS. The intensity above 0.3727 V/m is saturated for visualization.) As before, the intensity of the CLS method is too weak, while the MI method appears to activate the whole left hemisphere. Only SBFI provides a good balance between stimulation intensity, precision, and accuracy. The LCMV was unable to find a valid solution for the multi-target case. (**B**) The electric field distribution pattern of SBFI using a single $\lambda$ targeting synthetic multiple brain regions. (**C**) The electric field distribution pattern of SBFI using different $\lambda$.

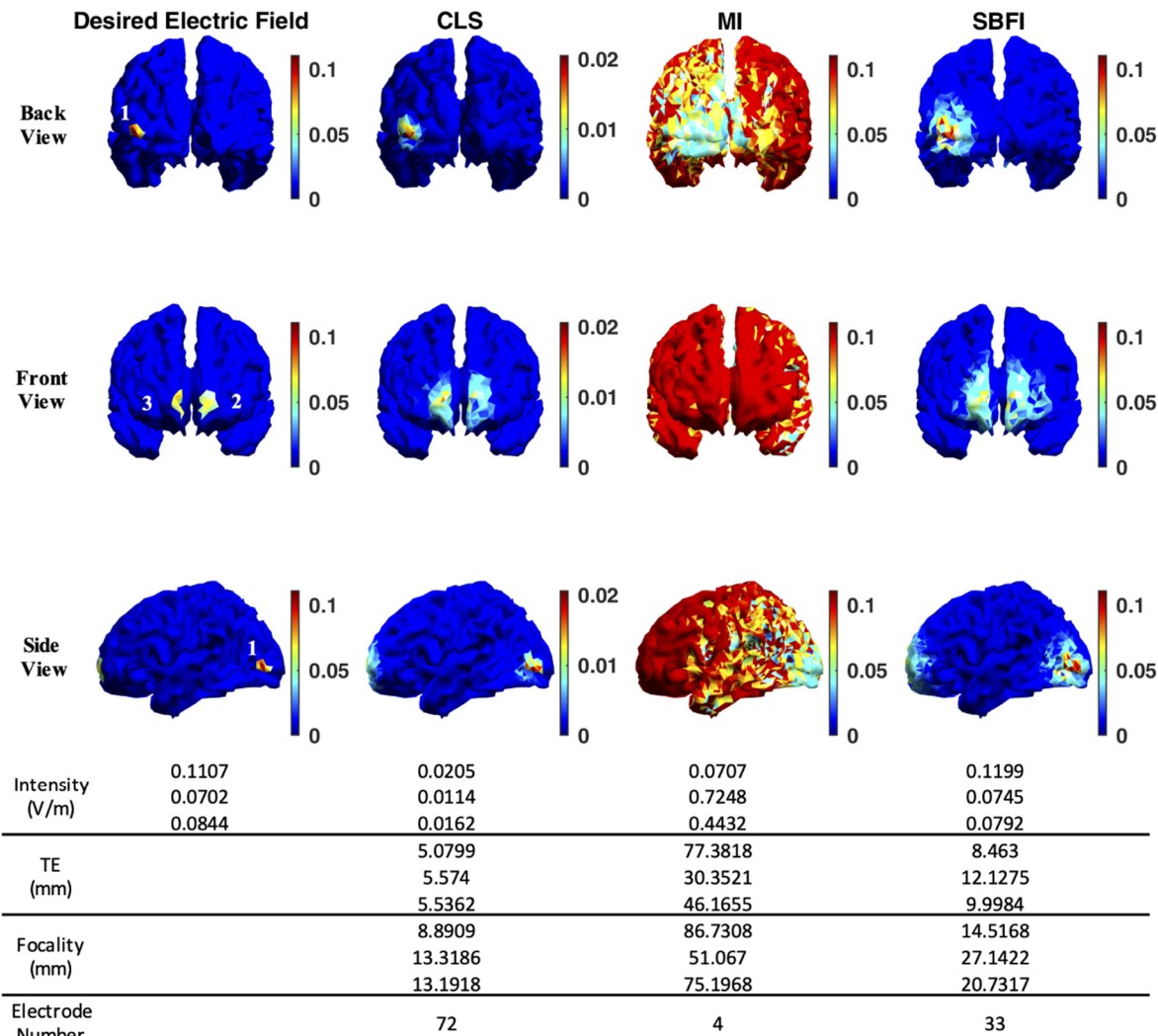

| | Desired Electric Field | CLS | MI | SBFI |
|---|---|---|---|---|
| Intensity (V/m) | 0.1107<br>0.0702<br>0.0844 | 0.0205<br>0.0114<br>0.0162 | 0.0707<br>0.7248<br>0.4432 | 0.1199<br>0.0745<br>0.0792 |
| TE (mm) | | 5.0799<br>5.574<br>5.5362 | 77.3818<br>30.3521<br>46.1655 | 8.463<br>12.1275<br>9.9984 |
| Focality (mm) | | 8.8909<br>13.3186<br>13.1918 | 86.7308<br>51.067<br>75.1968 | 14.5168<br>27.1422<br>20.7317 |
| Electrode Number | | 72 | 4 | 33 |

**Figure 3.** Results of different optimization algorithms for the real multiple targets protocol. The color scale represents intensity of the electric field, in V/m. (Note: Results are shown at the same scale [0, 0.1107] V/m for different algorithms except for CLS. The intensity above 0.1107 V/m is saturated for visualization.) Again, the intensity of the CLS method is too weak, while the MI method almost activates the whole brain. Only SBFI provides a good balance between stimulation intensity, precision, and accuracy. The LCMV method was unable to find a valid solution for the multi-target case.

In the real world, uncertainty in tissue conductivity can result in different electric field distribution from our expected, idealized models; this greatly affects the robustness of the computational model. The conductivity variations result in a different coefficient matrix $K'_{3n \times m}$; accordingly, with the same current pattern $s_{m \times 1}$, it produces an electric field vector $e'_{3n \times 1} = K'_{3n \times m} \cdot s_{m \times 1}$ that is different from the expected electric field distribution using standard conductivity values. The dissimilarity between $e'_{3n \times 1}$ and the expected distribution $e_{3n \times 1}$ reflects the robustness of the computational model. In this study, we observe how small changes in tissue conductivity can create differences between $e'_{3n \times 1}$ and $e_{3n \times 1}$, thereby analyzing the robustness of our model. Based on the literature [37–39], we assumed that the conductivities of the main tissues have uniform distributions with a unit of S/m, i.e., $P_{scalp}(x|\sigma) \sim U(0.2, 0.6)$, $P_{skull}(x|\sigma) \sim U(0.001, 0.04)$, $P_{CSF}(x|\sigma) \sim U(1.20, 2.01)$, and $P_{cortex}(x|\sigma) \sim U(0.05, 0.71)$. An overview of both single and multiple targets with different algorithms is given to demonstrate the general effects of conductivity changes. The detailed influences are revealed by an example of single target stimulation with SBFI in the main text.

### 2.5. Evaluation Metrics

Quantitative evaluations are used to measure and compare the performance of the proposed method with other state-of-the-art methods. At first, the stimulation intensity, in volts per meter, is quantified by the maximum electric field of the target region. In general, higher intensity is preferred. An intensity of 0.1~0.3 V/m is reported to be efficient [20,40,41].

$$E = e_{max} = max(e_{tar}) \tag{7}$$

Second, the target error (TE) is defined as the Euclidean distance between the mass centers of the target and the solution regions [18,22].

$$TE = \parallel MC_0 - MC \parallel_2, \quad (MC_0)_j = \frac{\sum_i (e_0)_{ij} \cdot p_{ij}}{\sum_i (e_0)_{ij}}, \quad (MC)_j = \frac{\sum_i (e)_{ij} \cdot p_{ij}}{\sum_i (e)_{ij}} \tag{8}$$

where $j \in \{x, y, z\}$, $p_i$ represents the coordinates of the $i$th voxel. The mass center of the target region and the activation region are $MC_0$ and $MC$, respectively. With units of millimeters, TE is a way of evaluating the stimulation accuracy: the smaller the TE, the higher the accuracy.

Third is the focality [18,22] measured in millimeters, which is represented by the radius in which the cumulative energy is half of the total energy. If we use $\Gamma(r)$ to represent the voxel set within a distance $r$ from the center of the target region, and $E(r)$ is the portion of the energy of $\Gamma(r)$.

$$r_{0.5} = r|_{E(r)=0.5}, \quad E(r) = \frac{\sum_{i \in \Gamma(r)} \parallel e(r_i) \parallel_2^2}{\sum_i e(r_i)_2^2} \tag{9}$$

Focality indicates stimulation precision. Smaller values of $r_{0.5}$ indicate that most of the energy is concentrated in a smaller region, and thus off-target brain regions are less likely to be activated.

In addition, although all optimization algorithms are applied to the aforementioned 342-electrode system, any electrode whose absolute current is less than 1 µA is considered inactivated, and the number of activating electrodes in the montage is provided. To avoid any confusion, the "electrode number" for different optimization methods in the rest of paper refers to the number of electrodes with absolute current larger than 1 µA.

When measuring model robustness, the mean squared error (MSE), the common image measurement, is adopted to evaluate the dissimilarity between $e'_{3n \times 1}$ and $e_{3n \times 1}$. A higher MSE indicates that the electric field distribution is more sensitive to changes in conductivity, and that the computational model is therefore less robust with respect to conductivity uncertainty.

$$\text{MSE} = \frac{1}{n} \sum (e'_{3n \times 1} - e_{3n \times 1})^2 \tag{10}$$

Then the intensity ($E'$), target error ($TE'$) and focality ($focality'$) of $e'_{3n \times 1}$ are compared with those of $e_{3n \times 1}$. $\Delta E$, $\Delta TE$, and $\Delta focality$ are therefore obtained by $|E' - E|$, $|TE' - TE|$, and $|focality' - focality|$, respectively. Finally, the maximum rate of change will be given, defined by the maximum $\Delta E / \Delta \sigma$, $\Delta TE / \Delta \sigma$, or $\Delta focality / \Delta \sigma$ for the different metrics.

## 3. Results

### 3.1. Study with Single Target

The advantages of using multiple electrodes instead of the conventional system with two large pad electrodes can be demonstrated in the results of the electric field distribution with a single target in Figure 1A,B. Figure 1C shows the electrode configurations calculated by different algorithms. Note, the configuration only shows the electrodes whose current is larger than 1 µA. The conventional montage produces a maximum electric field of 0.4385 V/m with a focality of 63.5043 mm. The figure clearly indicates this spread-

out distribution not only activates the target in the motor cortex, but also the entire left hemisphere. A target error of 28.3738 mm shows that the conventional montage does not create an accurate stimulation either. The performance of the MI algorithm is similar to the conventional montage with even higher electric field intensity and worse focality and accuracy, which is as expected. The MI method maximizes the intensity at the target reaching 0.7449 V/mm with only four electrodes. Although its intensity is more than 1.5 times higher than conventional system, it affects larger brain regions, especially the frontal lobe. The focality and target error are 81.9083 mm and 29.1206 mm, respectively. The CLS method has the opposite results of the MI algorithm and conventional montage. CLS produces an extremely precise and accurate stimulation pattern among all methods with a focality of 10.8616 mm and target error less than 1 cm. However, the outcome intensity of CLS is around 17 times lower than the desired value. Such a low intensity may not be clinically efficacious. Additionally, this montage requires 80 electrodes, which introduces additional drawbacks compared to the other methods. As for the LCMV and the proposed SBFI method, Figure 1 shows that the performance of those two are comparable. Both successfully find a balance between MI and CLS algorithms with around 20 electrodes. The desired intensity 0.3 V/m at the target region is achieved, while the focality and target error are within control. The focality of LCMV is 25.2921 mm, and its target error is 9.6710 mm. The proposed SBFI method with $\lambda = 0.0316$ is slightly better at the focality of 22.1130 mm and target error of 7.3776 mm. Compared to the conventional montage and MI algorithm, these dramatic improvements in focality and target error indicate that more energy is focused in a smaller region surrounding the target point, which promises more effective and safe stimulation.

### 3.2. Study with Multiple Targets

When stimulating disjointed target regions, the conventional two-electrode montage is no longer feasible, and using a multiple-electrode system is the only reasonable approach. Thus, the multiple-target studies exclude the conventional system, and comparisons are made only between different optimization methods. It is worth pointing out that LCMV performs reasonably in the single target case, but due to the hard constraint, the LCMV method has no feasible solution set for multiple target cases tested in our study. Therefore, we were not able to compare the performance of LCMV with all the other methods. The failure to have a feasible solution is the key drawback of the LCMV method, and this motivates us to propose the SBFI model.

Desired electric field distribution of synthetic multiple targets is shown in Figure 2A. Similar to the results of the single target study, CLS can mimic the electric field distribution pattern of the desired result, except the intensities of all three regions are almost an order of magnitude smaller than the desired value. The effectiveness of stimulation is questionable at such low intensities. On the contrary, the MI method produces high enough stimulation intensity in those three areas but sacrifices the focality. From the distribution figure, it is hard to recognize three discrete areas; the MI method appears to activate the whole left hemisphere. In the case of the proposed SBFI method, effective and precise stimulation can be achieved simultaneously with $\lambda = diag(0.5885, 0.247, 0.247)$. The intensity of all three regions is higher than 0.1 V/m, which is enough to induce cortical changes as shown in the literature [20,40,41]. Although the focality values of the three regions are relatively higher than in CLS, the energy outside the target region is actually lower than the threshold, which in turn will not be able to activate neurons. Thus, it is an acceptable tradeoff to see this small increase in focality value. Overall, the proposed SBFI method provides a good balance between stimulation intensity, precision, and accuracy. It not only can provide sufficient stimulation intensity as performed in the MI method, but also is able to minimize target error and focality simultaneously.

Figure 2B shows the results of SBFI using a scalar $\lambda$. It is clear that the scalar $\lambda$ fails to balance the three target regions. The desired intensity of all three target regions is similar, but the result intensity at the motor cortex region is much lower than the other two

regions. It appears the system does not favor the targets on the motor cortex region. One possible explanation could be the size of the target region. Intuitively, it is easier for the system to stimulate a large region than a tiny region because fewer electrodes are needed to minimize the intensity at the surrounding non-target region. If the target size is defined as the maximal distance between the target voxels and target mass center, the size of the motor cortex target is 12.9444 mm, which is the smallest among the three, compared to 13.6425 mm on the occipital lobe and 17.7783 mm on the temporal lobe. Thus, in order to compensate this unbalanced distribution of size, we have to assign different $\lambda_i$ for different target regions.

The results in Figure 2A clearly show that the proposed SBFI method can achieve reasonable balance between the intensity and focality by fine-tuning the optimization parameter $\lambda = diag\left(\lambda_{motor}, \lambda_{occipital}, \lambda_{temporal}\right)$. This flexibility allows the system to achieve different optimal results tailored to various applications. Decreasing the values of $\lambda$ will increase the weight of the first term $\| Ds \|^2$, which controls focality. The result will always have better focality and lower target error but with relatively lower intensity. For example, when $\lambda$ is chosen to be $diag(0.1085, 0.055, 0.055)$, the algorithm favors focality. As shown in Figure 2C, the focality and target error are better than the case of SBFI in Figure 2A, but the intensities are lower. Compared to CLS, SBFI achieves the highest intensities at all three target regions, while at the same level of focality and target error. Mathematically, increasing the values of $\lambda$ allows the second term to dominate, and the cost function is closer to the MI method. When $\lambda$ approaches infinity, SBFI is equivalent to the MI algorithm. The first term leads the optimization if $\lambda$ is decreased, as the cost function mainly minimizes the energies of the non-target area.

Similar results are obtained for the EEG-guided multiple target study as a simple demonstration that our proposed method is applicable to solve clinical needs, as shown in Figure 3. CLS can preserve the distribution pattern, but always fails to achieve enough intensity. MI undoubtedly reaches desired intensity, but the high intensity is at the cost of stimulation focality and accuracy. This EEG-guided case shows that MI even impacts both hemispheres, which may be due to the fact that there are targets on both hemispheres. It is worth pointing out that CLS and MI produce two extremes: MI favors the intensity of the targets and CLS produces more focal stimulation. However, only SBFI can balance the trade-off between intensity and the focality of all target regions. By choosing the proper optimization parameters $\lambda = diag(0.147, 0.1049, 0.1049)$, SBFI achieves desired intensity at all three target regions with reasonable focality and target error.

### 3.3. Robustness Study

We investigated our model's robustness to changes in conductivity values, including scalp conductivity, skull conductivity, cerebrospinal fluid (CSF) conductivity, and brain cortex conductivity. Since the electrical properties of these tissues can vary slightly between patients, we use the term "conductivity uncertainty" to describe the small changes made to our model parameters during our robustness study.

### 3.3.1. The Overall Impact of Conductivity Uncertainty

Figure 4A and Table 1 show the maximum MSE due to conductivity uncertainties for each algorithm. For all cases, the MSE peak appears at the skull layer, and valley appears at the CSF layer. These accordant peaks and valleys indicate a predominant impact of skull conductivity on model robustness, and a minor impact of the CSF on model robustness. The scalp layer and cortex layer have similar in-between effects on MSE, which indicate the comparable and moderate influences of the scalp and cortex in general. Figure 4B shows an example of MSE analysis in a single-target experiment with our SBFI method. MSE increases as the model's conductivity values deviate from their idealized values, which creates a concave "V" shape for each tissue. Note that the concave "V" shape associated with the skull is much sharper and higher than the others—this shows that the model is most sensitive to small uncertainties or deviations in skull conductivity. The

observed trends are consistent with all stimulation scenarios regardless of target types and optimization algorithms (see Supplementary).

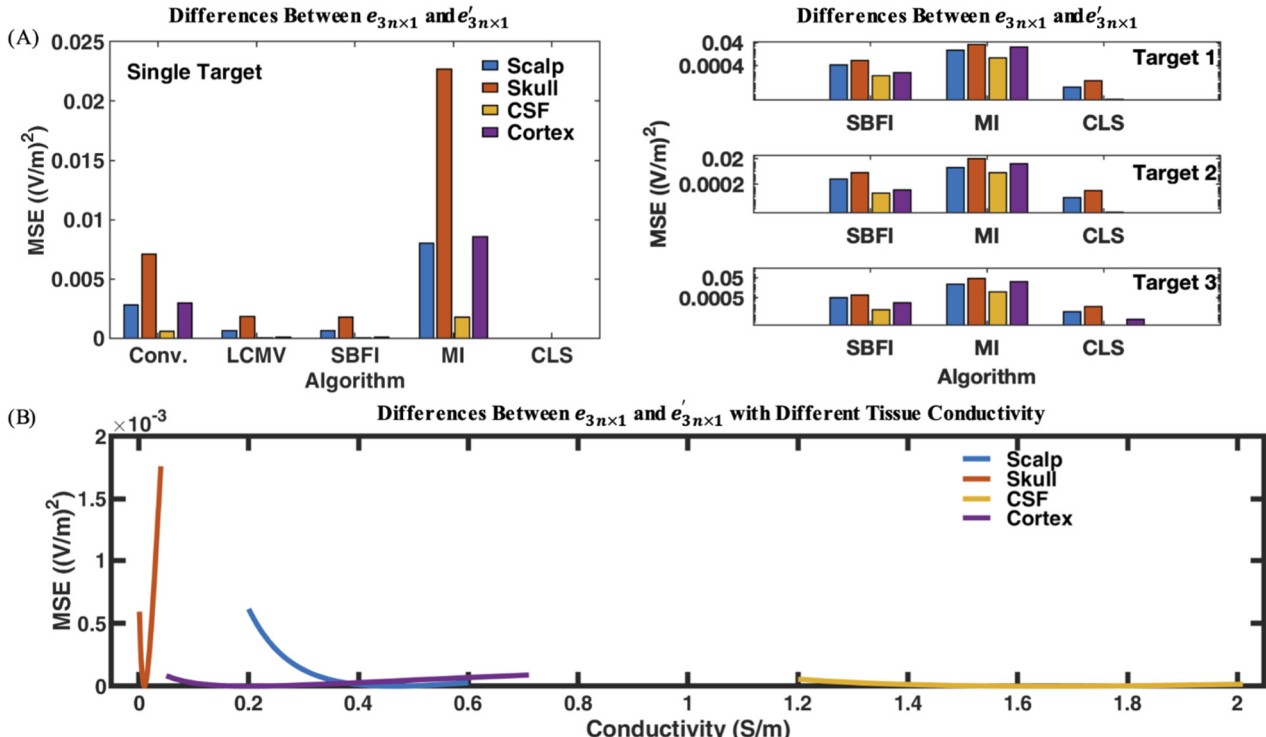

**Figure 4.** (**A**) Mean square error (MSE) between expected electric field $e_{3n\times1}$ and electric field $e'_{3n\times1}$ due to uncertainty in tissue conductivity. **Left**: single target. **Right**: synthetic multiple targets, from top to bottom are the results for target 1, 2, and 3, respectively. (Note: *y*-axis is log scale in the right column.) The effects on MSE in general: skull > scalp ≈ cortex > CSF. (**B**) Example of the MSE changes across the conductivity distribution range at each layer. When the conductivity of a tissue is equal to that tissue's original conductivity from the model, the MSE is trivially zero. However, as we shift the tissue's conductivity away from its starting value, the error increases. Larger slopes on the graph above indicate a high sensitivity to conductivity changes, and therefore a low robustness to uncertainty. Within the possible conductivity changes, the model is most sensitive to skull, and least sensitive to CSF.

**Table 1.** Dissimilarity results of the robustness test $((\text{V/m})^2)$.

| Tissue | Conventional | LCMV | SBFI | MI | CLS |
|---|---|---|---|---|---|
| Scalp | 0.003 | $6.43 \times 10^{-4}$ | $6.14 \times 10^{-4}$ | 0.008 | $6.79 \times 10^{-7}$ |
| Skull | 0.007 | 0.002 | 0.002 | 0.023 | $2.80 \times 10^{-6}$ |
| CSF | $5.71 \times 10^{-4}$ | $6.23 \times 10^{-5}$ | $5.51 \times 10^{-5}$ | 0.002 | $4.49 \times 10^{-8}$ |
| Cortex | 0.003 | $9.41 \times 10^{-5}$ | $8.89 \times 10^{-5}$ | 0.009 | $3.12 \times 10^{-8}$ |

### 3.3.2. The Impact on Intensity by Conductivity Uncertainty

Figure 5 and Table 2 show that the skull's conductivity uncertainty has the most powerful impact on electric field intensity. Figure 5B shows an example of intensity analysis in a single-target experiment with our SBFI method. An increase in skull conductivity induced a dramatic intensity growth on the target region. In contrast, the maximum intensity of the target region negatively correlates to the conductivity of the scalp, CSF, and cortex, on which the scalp has moderate effects stronger than the CSF and cortex. One possible explanation for the trends could be related to the conductivity ranges. The conductivity of the scalp/CSF is always much higher than the inner tissue layer skull/cortex. Therefore, the

current tends to be shunted through the layers of the scalp/CSF instead of the skull/cortex. If the scalp/CSF conductivity increases, current shunted though the scalp/CSF will increase, weakening the intensity at the cortex. If the skull conductivity increases, less current will be shunted at the scalp/skull boundary. At the skull/CSF boundary, the skull conductivity is still considerably lower than the CSF. The significant conductivity differences ensure an increase in the net current flowing into the CSF and cortex, resulting in an intensity increase at the cortex.

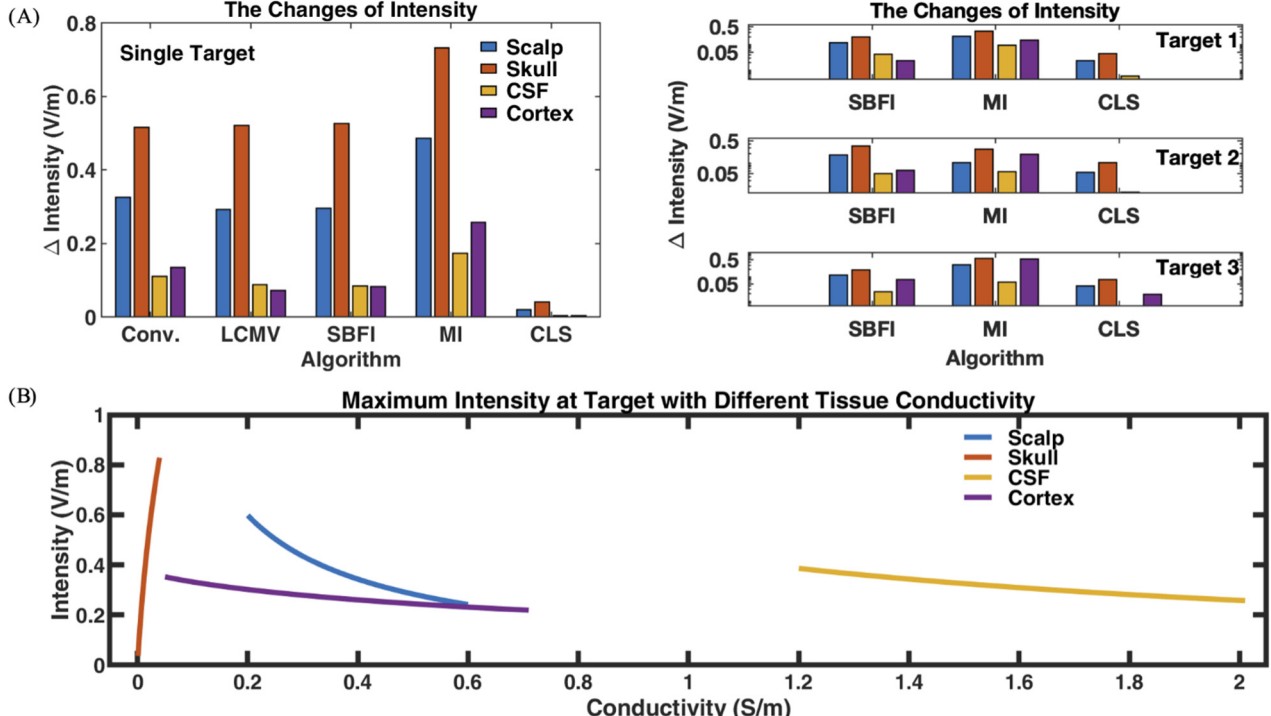

**Figure 5.** (**A**) Maximum intensity deviation from the expected value due to conductivity uncertainty. **Left**: single target. **Right**: synthetic multiple targets, from top to bottom are the results for target 1, 2, and 3, respectively. (Note: *y*-axis is log scale in the right column). The effects on intensity in general: skull > scalp > CSF ≈ cortex. (**B**) Example of the intensity changes. The large intensity spans on the graph above indicate the strong impact of the conductivity changes. Within the possible conductivity changes, skull conductivity increase greatly increases the intensity, while conductivity increases in scalp/CSF/cortex decrease the intensity.

**Table 2.** Intensity results of the robustness test.

| Tissue | $\sigma$ (S/m) | Max Intensity (V/m) | Min Intensity (V/m) | Rate ((V/m)/(S/m)) |
|--------|------------|---------------------|---------------------|--------------------|
| Scalp  | 0.2–0.6    | 0.598               | 0.242               | 0.890              |
| Skull  | 0.001–0.04 | 0.828               | 0.037               | 20.299             |
| CSF    | 1.20–2.01  | 0.386               | 0.257               | 0.159              |
| Cortex | 0.05–0.71  | 0.352               | 0.219               | 0.202              |

### 3.3.3. The Impact on TE by Conductivity Uncertainty

Regarding TE in Figure 6, the scalp and CSF generally have the least impact capped at 4 mm, which have no clear trend of increasing or decreasing for all scenarios (see Supplementary), while the skull and cortex can alter the TE greatly up to 16 mm. Figure 6B shows an example of TE analysis in a single-target experiment with our SBFI method. Overall, the TE tends to decrease when the skull/cortex conductivity increases. Sometimes, the cortex even induces more TE changes than the skull. However, considering the conductivity change of the skull is 94% smaller than the cortex ($\Delta\sigma_{skull} = 0.039$ S/m and

$\Delta\sigma_{cortex} = 0.66$ S/m), the rate of change of skull-induced TE is actually six times higher than that of the cortex, as shown in Figure 6B and Table 3. Thus, it is still reasonable to envision that the skull generally has stronger effects on TE than the cortex does.

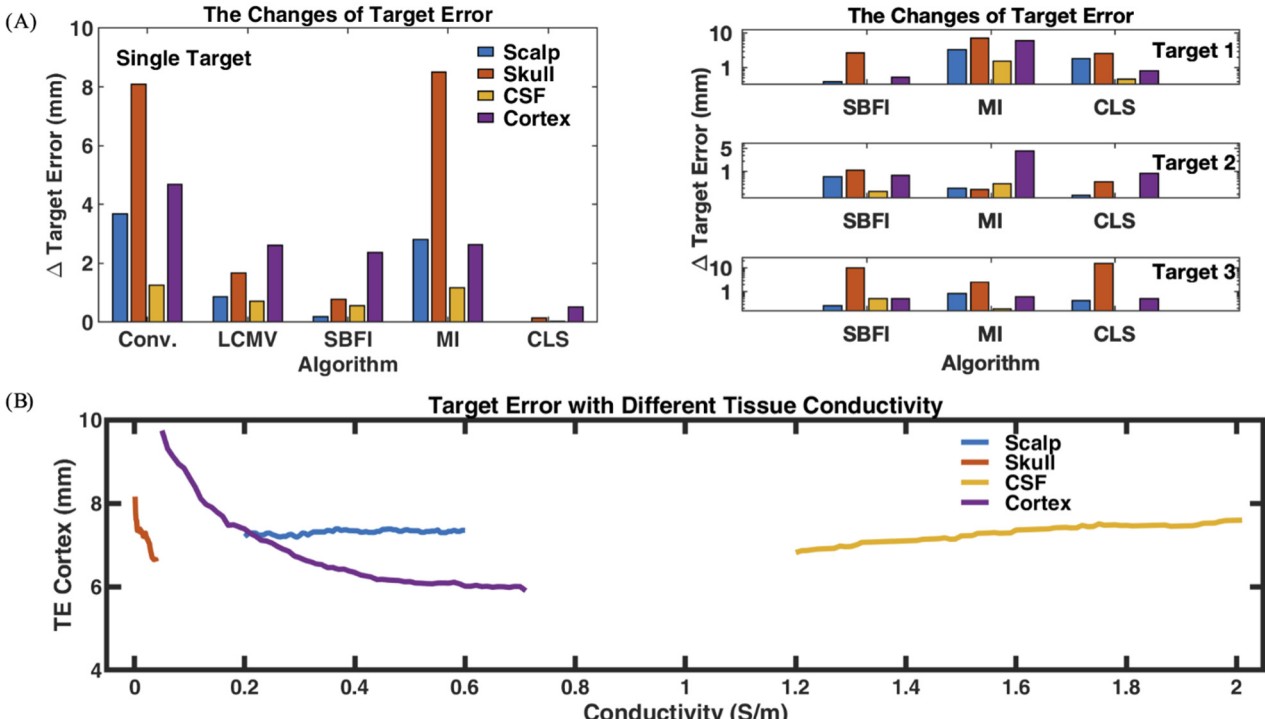

**Figure 6.** (**A**) Maximum TE deviation from the expected value due to conductivity uncertainty. **Left**: single target. **Right**: synthetic multiple targets, from top to bottom are the results for target 1, 2, and 3, respectively. (Note: *y*-axis is log scale in the right column.) The effects of TE in general: skull > cortex > scalp > CSF. (**B**) Example of the TE changes. TE has non-monotonic and divergent changing forms. (Note: the *y*-axis starts at 4 mm.) Overall, TE tends to decrease when the skull/cortex conductivity increases, while scalp/CSF has no clear tendency. TE spans caused by the cortex are the largest, which indicates that cortex conductivity has a strong impact on TE.

**Table 3.** Target error results of the robustness test.

| Tissue | $\sigma$ (S/m) | Max TE (mm) | Min TE (mm) | Rate ((mm)/(S/m)) |
|--------|-----------|-------------|-------------|-------------------|
| Scalp | 0.2–0.6 | 7.383 | 7.184 | 0.497 |
| Skull | 0.001–0.04 | 8.158 | 6.605 | 39.830 |
| CSF | 1.20–2.01 | 7.591 | 6.817 | 0.956 |
| Cortex | 0.05–0.71 | 9.743 | 5.900 | 5.822 |

### 3.3.4. The Impact on Focality by Conductivity Uncertainty

Similarly to the TE test, the scalp and CSF show minor and comparable effects on focality, usually inducing focality changes less than 10 mm. Figure 7A and Table 4 also reveal the main influence of skull conductivity uncertainty on focality. Figure 7B shows an example of focality analysis in a single-target experiment with our SBFI method. The skull-induced focality changes can be larger than 40 mm, but most of them happen when $\sigma_{skull} \leq 0.005$ S/m. When $\sigma_{skull} > 0.005$ S/m, the focality change is less than 1.5 mm. This also occurs in the other scenarios (see Supplementary), but the tuning point is not always the same. The impact of the cortex on focality is more complicated, and varies significantly with algorithms and target types. The highest change is $\sim 25$ mm, but sometimes the influence can be as weak as the scalp/CSF effects.

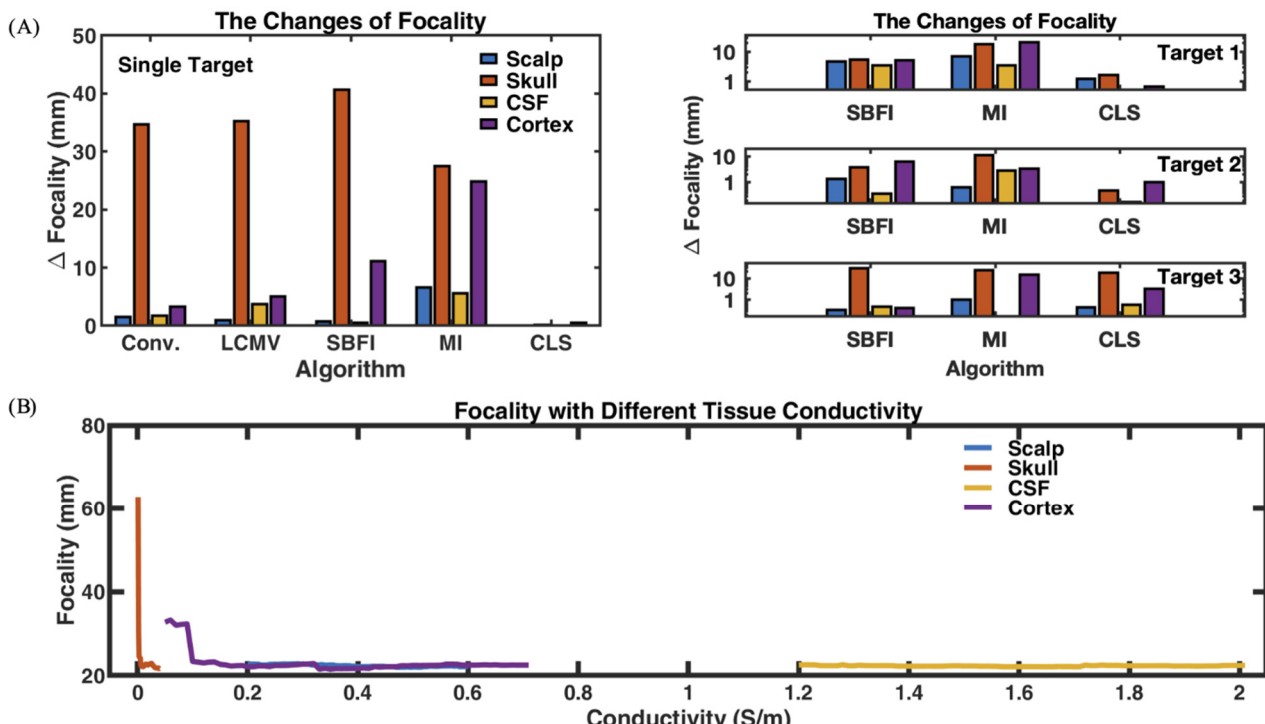

**Figure 7.** (**A**) Maximum focality deviation from the expected value due to the conductivity uncertainty. **Left**: single target. **Right**: synthetic multiple targets, from top to bottom are the results for target 1, 2, and 3, respectively. (Note: $y$-axis is log scale in the right column.) The effects on focality in general: skull > cortex > CSF ≈ scalp. (**B**) Example of the focality changes. (Note: the $y$-axis starts at 20 mm.) No clear tendency of the focality changes. Most focality changes happen when the skull conductivity is small.

**Table 4.** Focality results of the robustness test.

| Tissue | $\sigma$ (S/m) | Max Focality (mm) | Min Focality (mm) | Rate ((mm)/(S/m)) |
|--------|--------|-------------------|-------------------|-------------------|
| Scalp | 0.2–0.6 | 22.808 | 22.007 | 2.002 |
| Skull | 0.001–0.04 | 62.773 | 21.635 | 1054.8 |
| CSF | 1.20–2.01 | 22.598 | 22.102 | 0.612 |
| Cortex | 0.05–0.71 | 33.227 | 21.522 | 17.736 |

## 4. Discussion

### 4.1. Electrode Configuration

The electrode configuration for all studies clearly shows that MI always uses four electrodes, where two are used for stimulation and the other two are for current return. The two stimulation electrodes deliver most of the current to the targets to increase the intensity. However, with the safety constraint on the total amount of injected current, no spare electrodes can be used to neutralize the effects in non-target regions. This is why MI usually has poor focality and large target error. In contrast, CLS needs more electrodes than any of the other methods. These electrodes counteract each other to eliminate effects at non-target areas, which explains the good focality and small target error in all CLS results. The large number of electrodes also contributes to low intensity since each electrode can only deliver a small current to meet the safety constraint on total current. Therefore, the strategy is to keep the number of electrodes neither too large nor too small, which is the unique approach of SBFI. In a sense, some electrodes deliver enough current to ensure the desired intensity, while others are used to decrease the unwanted effects at non-target brain regions.

### 4.2. Regularization with Single Lambda and Multi-Lambda

As described in the results section, error in multiple-target studies can be mitigated by adopting $\lambda = diag(\lambda_1, \lambda_1, \ldots, \lambda_{n_{tar}})$ such that balancing each target can be performed simultaneously. The motivation to extend one scalar $\lambda$ to multiple $\lambda_i$ is that some targets may be in a dominant position and easier to be stimulated than others. For simplicity, we currently have all the voxels in the same region share the same $\lambda$. In an extreme case, we could set the optimal $\lambda_i$ for each voxel.

### 4.3. The Choice of Lambda

The selection of $\lambda$ strongly affects the outcome of intensity and focality. From the simulation study, we found that the optimal $\lambda$ varies from case to case, and thus there is no single fixed optimal value for every stimulation problem. Consequently, it is critical to decide the parameter $\lambda$ such that the balance between intensity and focusing ability can be obtained. The current selection is based on some sweep simulations and sophisticated methods, such as Bilevel Optimization [42,43] and Cross Validation [44,45], can be employed to choose $\lambda$ dynamically for closed-loop stimulations in the case of evolving sources. Nevertheless, insights could be obtained by studying the relationship among $\lambda$, the number of target voxels, and the size of target regions. For example, $\lambda$ is higher when stimulating a single target with only one voxel, and is much lower when stimulating multiple targets with hundreds of voxels. Thus, it is reasonable to predict that optimal $\lambda$ is inversely proportional to the number of target voxels, and adjusts $\lambda$ accordingly.

### 4.4. Robustness

The Robustness tests here identify the important roles of tissue conductivities in the optimal stimulation, which is consistent with the existing literature [37,38]. Through our systematic studies, we investigate the tissue conductivity variation effects for multi-electrode stimulations with different target types and different algorithms. Overall, scalp and CSF produce slight effects on stimulation intensity, target error, and focality. Unfortunately, conductivities of the skull and cortex itself greatly influence the electric field distribution over the cortex. These results indicate the need for individual modeling, especially for the parameter settings of the clinical applications. Another possible solution is to construct general models of specific populations. For example, the tissue conductivities of children, adults, and aging population vary a lot, while they may be stable within their own group [46–48]. Furthermore, conductivity discrepancies resulting from pathologies are also being investigated [49–51]. The electrical property changes in disease states may greatly influence the electric field distribution and alter the stimulation results. Thus, the specific simulation model for certain disease treatments should be carefully considered. As a result, more experiments should be conducted to investigate the joint influence of different tissue conductivity, various stimulation protocols, electrode characteristics, electrode position displacement, more precise human head models, etc.

It is interesting to note the robustness differences between algorithms that the MI method is in general less robust than other optimization algorithms. The conventional montage is slightly better than the MI method with relatively lower bars in Figures 4, 5, 6 and 7A, but the robustness is not as good as LCMV or SBFI. LCMV and SBFI have comparable robustness against tissue conductivity uncertainty. The reason for the more robust behavior could be that both algorithms try to balance both target ($e_0{}^T CS$) and non-target regions ($\| Ds \|^2$) with more electrodes. Therefore, the current distribution in the brain could be relatively more controllable, which leads to more robust results in the face of conductivity uncertainty. The results of CLS are the least affected by conductivity changes—far less than the others because of its low intensity over the whole cortex.

## 5. Conclusions

In this paper, we proposed a novel optimization algorithm: Stimulation with Balanced Focality and Intensity (SBFI) to support multiple electrode tDCS. SBFI can provide a balance

between stimulation intensity and focality by adjusting the optimization parameter $\lambda$ with a reasonable number of electrodes. Compared to the conventional montage and other popular optimization methods, SBFI can not only obtain sufficient stimulation intensity but also minimize target error and improve stimulation focality simultaneously. A series of simulation experiments present its potential for use in different clinical applications, especially stimulation for multiple targets. One limitation of the method is that the optimization parameter $\lambda$ is problem-dependent. In the discussions above, we outline possible solutions that will be explored in the near future. Furthermore, in the robustness studies, the proposed method SBFI shows a good robustness with different tissue conductivity variations. Among the tested algorithms, SBFI has relatively lower deviations from the overall electric field distribution and less intensity changes at the target area. The robustness experiments further suggest the high impact of skull conductivity variations, which requires more consideration in modeling studies and clinical implementation.

**Supplementary Materials:** The following supporting information can be downloaded at: https://www.mdpi.com/article/10.3390/a15050169/s1, Figure S1: Robustness test results-MSE; Figure S2: Robustness test results-Intensity; Figure S3: Robustness test results-TE; Figure S4: Robustness test results-Focality; Table S1: Synthetic Multiple targets dissimilarity results of the robustness test; Table S2: Synthetic Multiple targets intensity results of the robustness test; Table S3: Synthetic Multiple targets TE results of the robustness test; Table S4: Synthetic Multiple targets focality results of the robustness test.

**Author Contributions:** Conceptualization, W.L.; Data curation, Y.W.; Formal analysis, Y.W. and J.B.; Funding acquisition, W.L.; Investigation, Y.W.; Methodology, Y.W., and W.L.; Resources, W.L.; Supervision, W.L.; Validation, J.B.; Writing–original draft, Y.W.; Writing–review & editing, Y.W., J.B., and W.L. All authors have read and agreed to the published version of the manuscript.

**Funding:** The research is partially supported by the Bioengineering Department fellowships, Distinguished Professorship, Patrick Soon-Shiong Endowment Funds, and Chan Soon-Shiong Bionic Engineering Center at UCLA (Project Number: CNSI-2012-1184), all at UCLA.

**Institutional Review Board Statement:** Not applicable.

**Informed Consent Statement:** Not applicable.

**Data Availability Statement:** The data used to support this study are included within the article and its Supplementary material.

**Conflicts of Interest:** WL holds shareholder interest in Niche Biomedical Inc. The others declare that the research was conducted in the absence of any commercial or financial relationships that could be construed as a potential conflict of interest.

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
