# Peer review of "Stimulation Montage Achieves Balanced Focality and Intensity"

_algorithms, doi:10.3390/a15050169_

Round 1

Reviewer 1 Report

The work studies the optimal montage and optimization method in transcranial electric stimulation. The manuscript is well written and the work is pertinent and interesting. However, it is mandatory to the authors to consider the comments below sufficiently.

Comments:   

— line 105: the authors should clarify the notion of applied currents. In practice, electrodes are applied with two poles between which the current flows. Here, the number of electrodes is the number of currents,
what is not correct since already two electrodes  induce a single current. 

— line 125 : how did the authors choose the matrix C  ?

— line 141: how didi the authors choose the weights \lambda_i ?

— section 2.2: please clarify additionally which parameters are estimated and which are chosen a priorily. 

— Eq. (3): in the context of the current distribution, Eq.(3) is hard to understand and appears to be non-valid since tDCS in practice induces a non-vanishing current between electrodes. If  Eq.(3) is mandatory from a mathematical point of view, the authors should clarify how to implement this in practice. Especially the danger of short circuits might imposes some problems, please discuss this. 

— line 147: “Next is the maximum current injected into each electrode,… ” : rephrase 

— Eq.(5): explain why you introduced the factor 2.

— line 191: “First is the stimulation intensity in 191 volts per meter, ”: rephrase, e.g. like “At first, , the stimulation intensity….”

— line 216: define better Delta E, Delta TE , Delta TE , Delta focaility

- section 4.1: it is far from being clear why different optimization techniques estimate different number of electrodes, see also comment above. This has to be clarified.

- section 4.2: a Discussion section should not contain results, such as Fig. 8. This figure has to be moved to  and discussed in the Results section. In addition, results are given in this section and they have to be moved to the Results section .
- section 4.3 and Fig.9: see the comments to section 4.2 .

Reviewer 2 Report

The paper by Wang, et al. presents a new algorithm for achieving focal tDCS which balances between the need to achieve a minimum intensity in one or more target regions with the need to achieve focality, overcoming some limitations of previous algorithms applied to this task. The work presented in the paper is adequately well presented and appears to be fundamentally sound, if lacking in real-world validation. However, some improvements could be made.

Line 36, can the authors present a 1-sentence summary to specify how the reciprocity theorem has been applied previously?

Line 61, What three parameters does L1 normalisation introduce?

Equation 1, the description of this equation is a little unclear to me, isn't C a submatrix of K? what are dimensions of e0 and C? Adding these clarifications will help clarify the expansion to the multi target case as well.

Line 162, how were the target values for the E field in the different areas of the brain chosen? Again for the EEG case, unclear why this mentioned as no EEG source localisation is presented in the scope of the current paper.

Line 172, it is worth mentioning here that the comparison is therefore between SBFI, MI, and CLS.

Line 180, have you considered electrode position effects on error? This is a very common source of error for real clinical applications especially with EEG as opposed to ECoG type approaches. Probably beyond the scope of the current paper but worth mentioning in the discussion.

Figure 1,

- "A", "B", "C" labels for the different rows would be helpful.

- For the bottom row, varying colour scale makes it very difficult to compare across methods, please use same and symmetric-about-zero scale. For the top rows this can be forgiven in CLS E-field case because would be illegible and the remainder appear to be scaled in the interval [0, 0.3]. This doesn’t appear to match the intensity numbers in table 1 however.

- As claim about intensity and focality is made in caption, please be clear to link this interpretation of the figures to the numbers from table 1. These could even be added to the figure on the images in the second row, more clearly linking the metrics to the data and removing the need for a separate table (similar comment applies to the other tables and figures).

Line 247, “feasible” … do you mean clinically efficacious?

Figure 2, It would be good to move the results from section 4.2 up to this figure as I really wanted to know how much these results depended on lambda and what techniques you used to pick lambda here (and section 4.2 is much more results than discussion). Figures 2, 8, and 9 could all be merged (and figures 8 and 9 presented on a consistent colour scale).

Figures 4-7, I found the right column of panel A squished and difficult to read especially given the varying scales … is there a way to re-arrange your presentation to emphasise legibility? Log Y scales might greatly help. It’s unclear what configuration (mono vs. multifocal 1 or 2) is used for panel B.

Figure 6, I have a little difficulty reconciling the relatively small red curve in 6B with the large delta errors for skull in 6A across the different approaches

Figure 6b, 7b. Please start the graph Y axis at 0. This is necessary for correct graphical interpretation of results.

Line 389, This is an unnecessary contortion to force the effect of the skull to always be greatest in theory. It’s OK for that to not be the case for every metric and configuration, the results are what they are! Please don’t twist your results to try to achieve flawless consistency.

Line 464, when presenting lambda vectors in the text it would be helpful to remind the reader what the order of elements is in correspondence to the brain anatomy (e.g. motor, pariatal, temporal?). Could also be labelled on figures which entry corresponds to where.

Line 471, and as L approaches 0 the equation approaches ... LCMV?

Round 2

Reviewer 1 Report

The authors have addressed a large part of the reviewers comments. However, they should address the comment given below, that has already been mentioned in the previous report:

line 131/132: the statement is not correct, since \lambda has to be optimized as well and this has to be stated clearly in the Methods section. In this context, the majority of section 4.3 has to be moved to the Methods section explaining the choice of \lambda.
